# The mesh is a network of microtubule connectors that stabilizes individual kinetochore fibers of the mitotic spindle

Faye M Nixon[1,2], Cristina Gutiérrez-Caballero[1], Fiona E Hood[2], Daniel G Booth[2†], Ian A Prior[2], Stephen J Royle[1*]

[1]Division of Biomedical Cell Biology, Warwick Medical School, Coventry, United Kingdom; [2]Department of Cellular and Molecular Physiology, Institute of Translational Medicine, University of Liverpool, Liverpool, United Kingdom

**Abstract** Kinetochore fibers (K-fibers) of the mitotic spindle are force-generating units that power chromosome movement during mitosis. K-fibers are composed of many microtubules that are held together throughout their length. Here, we show, using 3D electron microscopy, that K-fiber microtubules (MTs) are connected by a network of MT connectors. We term this network 'the mesh'. The K-fiber mesh is made of linked multipolar connectors. Each connector has up to four struts, so that a single connector can link up to four MTs. Molecular manipulation of the mesh by overexpression of TACC3 causes disorganization of the K-fiber MTs. Optimal stabilization of K-fibers by the mesh is required for normal progression through mitosis. We propose that the mesh stabilizes K-fibers by pulling MTs together and thereby maintaining the integrity of the fiber. Our work thus identifies the K-fiber meshwork of linked multipolar connectors as a key integrator and determinant of K-fiber structure and function.

*For correspondence: s.j.royle@warwick.ac.uk

Present address: †Wellcome Trust Centre for Cell Biolog, University of Edinburgh, Edinburgh, United Kingdom

Competing interests: The authors declare that no competing interests exist.

## Introduction

Accurate mitosis is essential to eukaryotic life. It requires the correct assembly of a bipolar array of microtubules (MTs) into a mitotic spindle which, in concert with hundreds of different motors and non-motor proteins, segregates the duplicated sister chromatids to the two daughter cells. Many of the chromosome movements in mitosis are governed by the kinetochore fibers (K-fibers) of the spindle apparatus. In human cells, K-fibers are bundles of 20–40 parallel MTs that typically run from the kinetochore to the spindle pole (*McDonald et al., 1992*; *Mastronarde et al., 1993*; *McEwen et al., 1997*; *Booth et al., 2011*; *Sikirzhytski et al., 2014*). K-fibers can be thought of as coherent units: their constituent MTs are held together throughout their length, as well as being focused at either end (*Rieder, 1981*; *Spurck et al., 1997*). The coherence of the K-fiber is thought to be crucial for accurate chromosome segregation in mitosis. However, the ultrastructural and molecular basis of K-fiber coherence is not well understood.

Lateral MT connectors are important for the function of MT arrays, such as K-fibers, in cells (*Brangwynne et al., 2006*; *Ward et al., 2014*). Classic electron microscopy (EM) studies uncovered the presence of electron density between MTs of the K-fibers (*Wilson, 1969*; *Hepler et al., 1970*; *Witt et al., 1981*; *Bastmeyer and Fuge, 1986*). These inter-MT bridges appear as bipolar struts laterally connecting two MTs in 2D electron micrographs. The bridges are typically ~5 nm thick and range from 6–20 nm in length. The morphology of bridges is heterogeneous, and they are likely composed of a variety of proteins (*Booth et al., 2011*). The mitotic spindle is an ensemble of hundreds of MT-associated proteins (*Sauer et al., 2005*), some of which are candidates for inter-MT bridges (*Manning and Compton, 2008*). The number of inter-MT bridges scales with the number of

**eLife digest** Before a cell divides, its genetic material must be copied and then equally distributed between the newly formed daughter cells. In the cells of plants, animals, and fungi, a structure known as the spindle pulls the two copies of the chromosomes apart. The spindle is made up of a network of long, protein filaments called microtubules, and the bundles of microtubules that attach to the chromosomes are referred to as 'K-fibers'.

K-fibers are organized in a way that provides strength. These bundles of microtubules are held together throughout their entire length and, in 2011, it was suggested that a group of proteins including one called TACC3 could cross-link adjacent microtubules within K-fibers. However, it remained unclear how these proteins achieved this.

Now, Nixon et al.—including several of the researchers involved in the 2011 work—have used a technique called 3D electron tomography to analyze what holds the K-fibers together in human cells. This analysis revealed struts or connectors that hold together adjacent microtubules within K-fibers. These connectors can vary in size and a single connector can link up to four microtubules. This means that, in a three-dimensional view, the connectors appear as a 'mesh' between the microtubules in the bundle.

Nixon et al. then increased the levels of the TACC3 protein and found that the K-fibers became disorganized. The spacing of the microtubules with the K-fibers was reduced so that they were more tightly packed than normal. These observations suggest that 'the mesh' influences the microtubule spacing within a K-fiber.

Nixon et al. analyzed how disorganized K-fibers affected dividing cells and found that it took longer for the chromosomes to move to the newly forming daughter cells. This suggests that cells must maintain optimal levels of TACC3 to ensure that the K-fibers can effectively separate the chromosomes. Further work is needed to identify the other proteins and molecules that make up the mesh.

MTs and also with the number of paired MTs in a bundle (*Bastmeyer and Fuge, 1986*), and it was proposed that it is the inter-MT bridges that hold K-fiber MTs together (*Witt et al., 1981*). The connection between the kinetochore and K-fiber MTs has been studied by 3D-EM (*Dong et al., 2007*; *McIntosh et al., 2008*). However, a similar 3D investigation of inter-MT connections away from the kinetochore has not been reported.

One molecular candidate for inter-MT bridges in K-fibers is TACC3–ch-TOG–clathrin. The assembly of this complex is regulated by phosphorylation of TACC3 at serine 558 by Aurora-A kinase (*Fu et al., 2010*; *Hubner et al., 2010*; *Lin et al., 2010*; *Booth et al., 2011*). This allows two domains from TACC3 and clathrin to come together in space, making a single, composite MT interaction surface (*Booth et al., 2011*; *Cheeseman et al., 2011*; *Hood et al., 2013*). Previous work showed that this complex is necessary for K-fiber stabilization (*Royle et al., 2005*; *Booth et al., 2011*; *Cheeseman et al., 2013*) and that the complex forms a distinct class of short inter-MT bridges (*Booth et al., 2011*).

In this study, we set out to examine the 3D ultrastructure of K-fiber inter-MT bridges. Surprisingly, we found that inter-MT bridges are not simply bipolar connections between two MTs but are a network of interconnected struts that contact multiple MTs. We term this structure 'the mesh'. This novel subcellular structure positions MTs within the K-fiber and is required for normal mitosis.

## Results

### Inter-MT connectors in K-fibers are 'bridges' in 2D and a 'mesh' in 3D

Inter-MT bridges were first described in 2D electron micrographs as faintly stained fine threads projecting from the surface of the tubules (*Hepler et al., 1970*). For example, 2D views of inter-MT bridges connecting adjacent MTs within a K-fiber are shown in *Figure 1A*. The morphology of inter-MT bridges is heterogeneous (*Booth et al., 2011*), as noted in the original report (*Hepler et al., 1970*). In order to examine the morphology of bridges in more detail, we used electron tomography of sections taken from mitotic HeLa cells fixed by high-pressure freezing/freeze substitution (HPF/FS). This gave a 3D view of the K-fiber MTs and the material that connected them (*Figure 1B*, *Video 1*). Careful examination of the

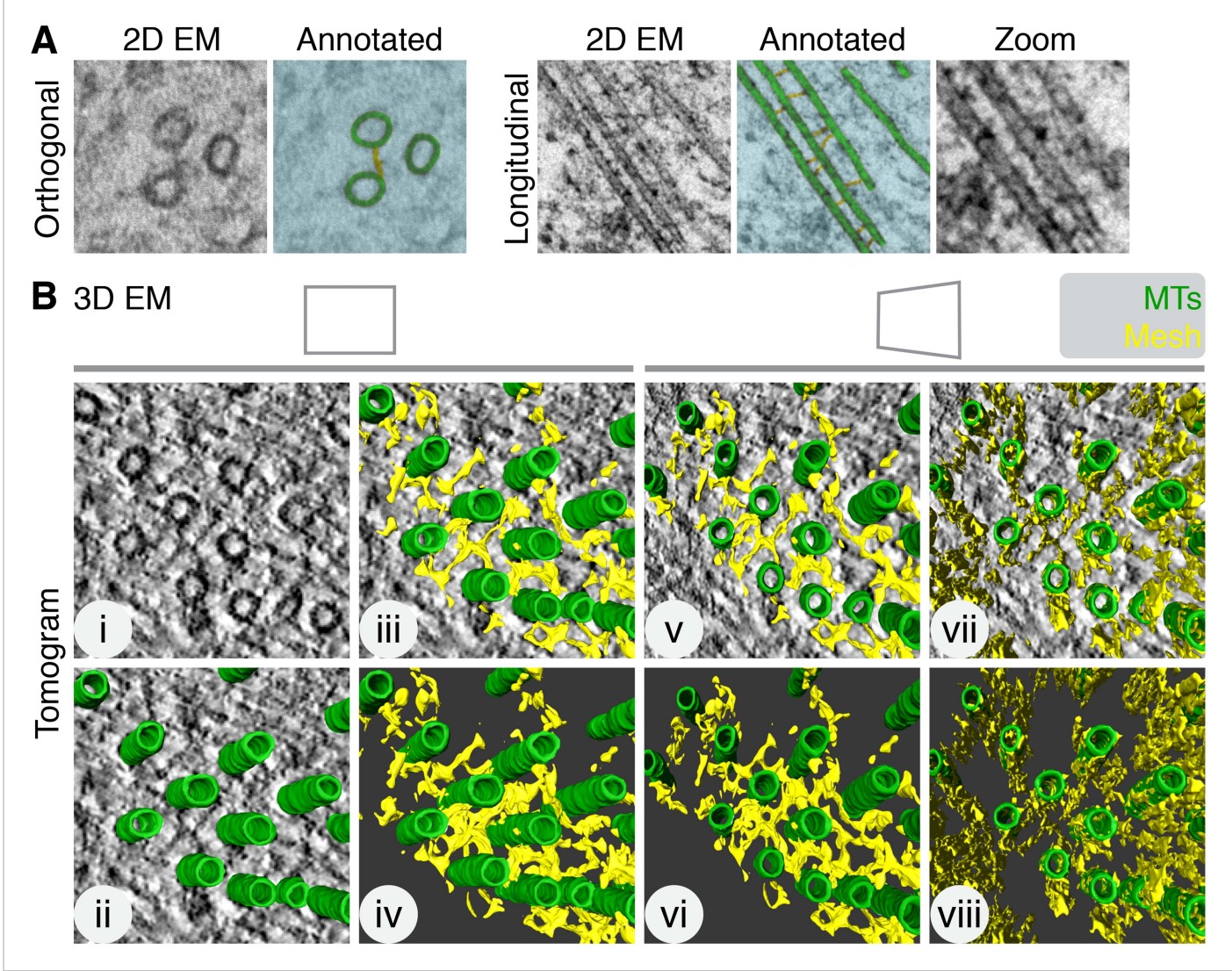

**Figure 1**. Intermicrotubule connectors in K-fibers are 'bridges' in 2D and a 'mesh' in 3D. (**A**) 2D views of inter-MT bridges in sections taken orthogonally or longitudinally to the spindle axis. In the annotated version, MTs (green) and inter-MT bridges (yellow) are shown on a blue background. Zoom of longitudinal section is a 2× expansion of the lower part of the 2D EM view. (**B**) Orthoslice of a tomogram generated from a tilt series of a single section through a K-fiber preserved using HPF/FS (**i**). Overlaid is a hand-rendered 3D representation of MTs (green) (**ii**) and associated mesh (yellow) (**iii**). The model is shown alone (**iv**). For (**v**–**viii**), the tomogram is tilted to show the MT axis. In **v** and **vi**, the overlay and model are shown. In **vii** and **viii**, the same view is shown but with the mesh detected by the automated segmentation method. Note that sections taken >1 μm away from the kinetochore are shown in this and all subsequent figures. Tomogram thickness, 45.6 nm. For scale, MTs are 25 nm in diameter.

The following figure supplements are available for figure 1:

**Figure supplement 1**. The mesh is visible, but not well preserved, in chemically-fixed K-fibers.

**Figure supplement 2**. The mesh is associated with K-fiber MTs.

tomograms revealed that inter-MT bridges are not just simple bipolar struts connecting two adjacent MTs: they are interconnected and can contact multiple MTs within the K-fiber. This network of inter-MT connectors is best illustrated by rendering the densities seen in the tomogram and visualizing the resultant computer model (*Figure 1B*). We term the interconnecting material 'the mesh'.

To obtain an unbiased view of the mesh, we developed a semi-automated segmentation method for 3D model building (*Figure 1B*, *Video 1*). This method was used for all subsequent quantification.

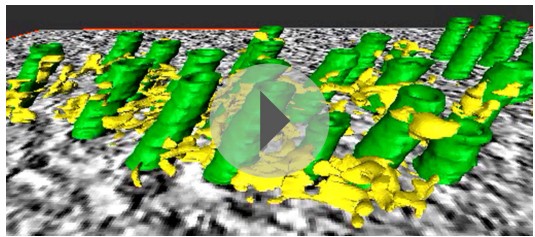

**Video 1.** Example of the K-fiber mesh from a normal HeLa cell at metaphase. Tomogram of a K-fiber. The mesh (yellow) is shown by manual rendering and by automated rendering. MTs (green) were rendered by hand. All segmentation was smoothed in Amira.

We found that mesh preservation was superior using HPF/FS compared with chemical fixation (*Figure 1—figure supplement 1*). Moreover, the mesh was associated with K-fiber MTs. In contrast, segmentation of non-K-fiber areas resulted in distinct, particulate density and translating MTs to a non-K-fiber area meant that globular density, which passed through the MTs, was detected (*Figure 1—figure supplement 2*).

## Ultrastructural morphology of the mesh

Having established a detection method, we next explored the anatomy of this novel subcellular structure. Three types of connectors within the mesh could be distinguished in these 3D models. Bipolar, tripolar, or quadrupolar connectors could be selected using the criterion of uninterrupted density that connects two, three, or four MTs, respectively (*Figure 2A*). These connectors were highly heterogeneous in size. Where the volumes of individual connectors could be determined easily, we found that bipolar, tripolar, and quadrupolar connectors had mean volumes of 6,689, 17,370, and 32,376 nm$^3$, respectively (*Figure 2B*). Therefore, there is not a linear relationship between the number of struts in a connector and the connector volume. The volume of mesh within each fiber was, on average, equivalent to the volume of MTs that it encapsulates and larger than the volume of MT walls alone (*Figure 2C*). This observation establishes that the mesh is a major component of every K-fiber.

We next examined the contacts made between the mesh and MTs to determine if there were preferred locations on the MT for mesh attachment, for example, on the seams of MTs. We found that mesh-MT contacts were highly heterogeneous (*Figure 2D*). For example, a crosslink between two MTs could be a simple bipolar strut with a small footprint on both connected MTs, or at the other extreme, the crosslink could be a network of multiple-linked struts, with large mesh contact areas on both MT surfaces. These larger, composite footprints extended for some distance along both linked MTs, but these were not confined to a co-axial line on the MT wall. Moreover, several mesh connectors could contact the same MT at the same axial position (e.g., *Figure 1B*), which further argues that the mesh has no preference for the MT seam.

## Manipulating the mesh influences K-fiber organization

Previous work indicated that one component of the mesh is a complex of TACC3–ch-TOG–clathrin whose assembly is regulated by Aurora-A kinase (*Booth et al., 2011*; *Hood et al., 2013*). Moreover, the amount of this complex on K-fibers can be increased simply by overexpressing TACC3 (*Booth et al., 2011*). In order to experimentally manipulate the mesh, we therefore made a stable inducible HeLa cell line where GFP-TACC3 could be overexpressed in a controlled manner (*Figure 3—figure supplement 1*). The most obvious effect of this manipulation was to alter MT organization. K-fibers in cells expressing GFP-TACC3 had more MTs per fiber, and the cross-sectional area that those MTs occupied was larger, compared to fibers in control uninduced HeLa cells (*Figure 3A*). The MT density was similar between the two groups suggesting that the fiber area scaled with the number of MTs (*Figure 3A*).

To look in more detail at MT packing within a K-fiber, we analyzed the distance from each MT to its nearest neighbor, and the number of neighboring MTs found within 80 nm. Both analyses showed that *local* MT packing density within the fiber had increased substantially in TACC3 overexpressing cells compared to uninduced controls (*Figure 3B*), although the fibers themselves were larger overall. A simple manifestation of this tighter local packing was the increased frequency of doublet and triplet MTs within TACC3 overexpressing K-fibers (*Figure 3B*). The median distance to the nearest neighboring MT had decreased from 56.1 to 48.1 nm (*Figure 3C*), a change in edge-to-edge proximity from 31.1 to 23.1 nm. This means that in TACC3 overexpressing cells, the average nearest neighboring MT is less than the width of one MT away. Because the overexpression of TACC3 alters the MT packing within the K-fiber, these experiments suggested to us that the mesh might influence

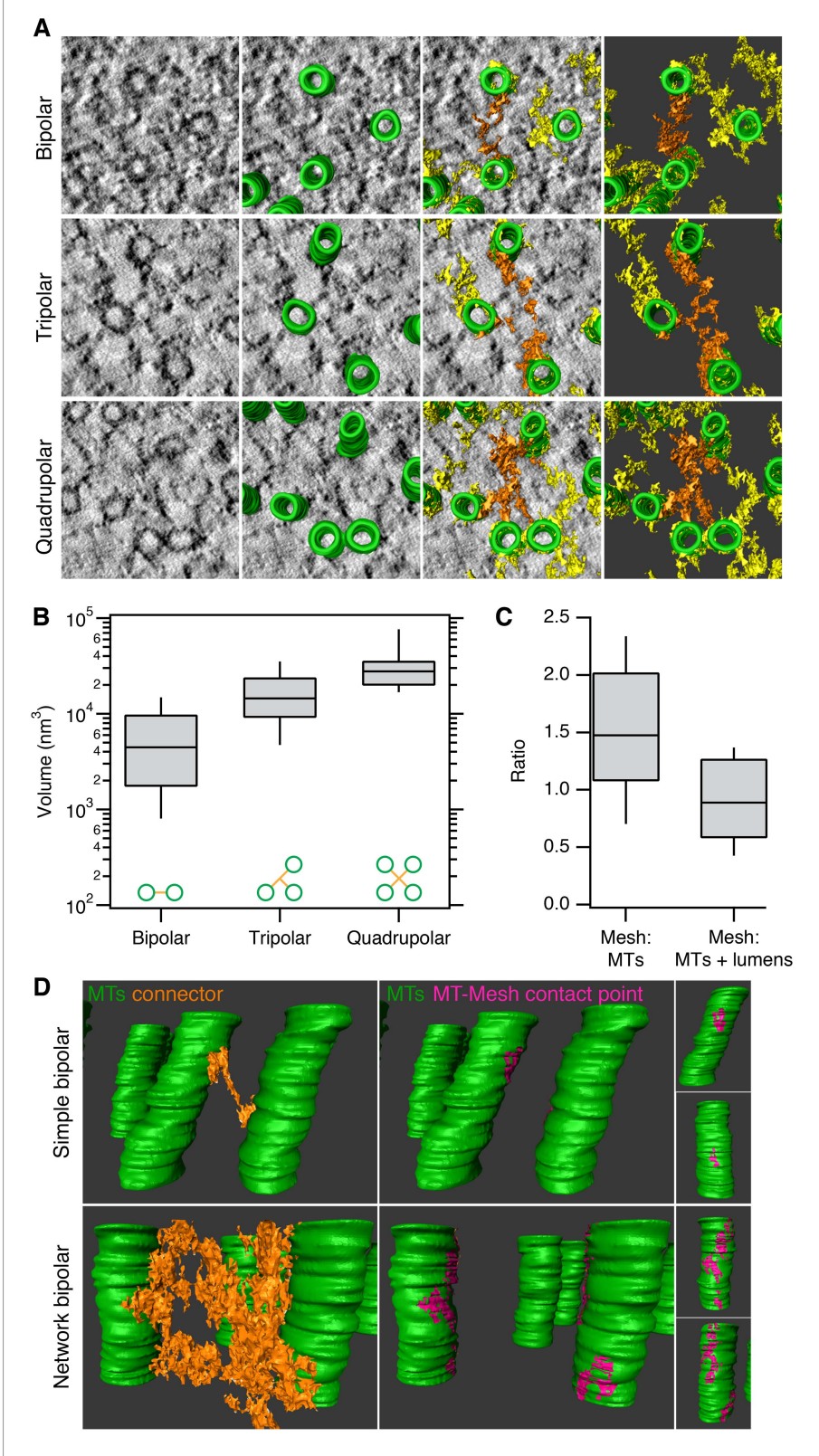

**Figure 2**. Inter-MT linkages are defined connectors with heterogeneous MT-mesh contacts. (**A**) Examples of bipolar, tripolar, and quadrupolar connectors within the mesh. Single orthoslices from tomograms showing examples of different connectors. MTs are hand-rendered (green), mesh is automatically detected (yellow, see 'Materials and methods'), and example connectors are highlighted orange. Tomogram thickness, 51.3 nm (bipolar) and
*Figure 2. continued on next page*

*Figure 2. Continued*

35.2 nm (tripolar and quadrupolar). (**B**) Box plots to show the volume of bipolar, tripolar, and quadrupolar connectors within the mesh from multiple tomograms. (**C**) Box plot to show the ratio of the volume of mesh relative to MTs (walls only) or MTs + lumens ('filled-in' MTs). Box plots show the median, 75th and 25th percentile, and whiskers show 90th and 10th percentile. (**D**) Heterogeneity of contacts between mesh and MTs. Two pairs of MTs are shown with a simple bipolar connector (above) or with a more complex connection (below). MTs (green) are shown with a single component of the mesh (orange). The contact points between the selected mesh and the MTs are shown in pink. Note the extensive mesh-MT contacts in the lower example. Tomogram thickness, 66.4 nm (simple) 64.8 nm (network). For scale, MTs are 25 nm in diameter.

MT spacing within the K-fiber. We return to the hypothesis that the mesh has an important role in MT spacing below.

Are the additional MTs in TACC3 overexpressing K-fibers stably attached to the kinetochore? To address this question, we used a 3D confocal microscopy assay of tubulin staining in the vicinity of kinetochores (*Cheeseman et al., 2013*). In agreement with the EM analysis, we detected a higher tubulin signal in cells expressing GFP-TACC3 compared to those expressing GFP alone. Following cold treatment, the tubulin intensity in the vicinity of kinetochores was reduced to comparable levels in both conditions, suggesting that the additional MTs in TACC3 overexpressing K-fibers are attached by mesh to the rest of the K-fiber but were not stably attached to the kinetochore (*Figure 3D*).

Overexpression of TACC3 also increased the volume of mesh between K-fiber MTs in a tomogram to $5.2 \pm 1.0 \times 10^6$ nm$^3$ (mean $\pm$ s.e.m.). This corresponds to $9.1 \pm 0.01\%$ of the fiber volume in the tomogram, whereas control mesh was $5.7 \pm 0.01\%$. This change is somewhat difficult to interpret because of the significant increase in the number of MTs per fiber and the tighter local packing. More MTs per fiber might push up the volume of mesh, but the closer proximity of MTs limits the space available for mesh to be present.

## Overexpression of TACC3 increases MT interconnectivity in K-fibers

One defining characteristic of the mesh is that it connects multiple MTs within K-fibers. This interconnectivity means that a MT that is contacted by the mesh is connected to one or more MTs and each of these, in turn, may be connected to one or more MTs and so on. We defined these connected MTs as 'chains'. In uninduced cells, chain sizes were small, containing at most 6 MTs (*Figure 4A*). By contrast, cells overexpressing TACC3 had chains containing up to 12 MTs (*Figure 4A*), suggesting that the MTs were more interconnected as a result of TACC3 overexpression. Although MTs in TACC3 overexpressing K-fibers were more interconnected, the constitution of the connectors within the mesh was not noticeably altered (*Figure 4B*). In both conditions, the mesh was composed of a predominance of bipolar connectors and similar proportions of tripolar and quadrupolar connectors (*Figure 4B*).

We next wondered if the larger chains in TACC3 overexpressing cells were the result of the tighter local MT packing. Accordingly, we constructed 2D MT maps, where each MT's chain membership was displayed, and we compared these to the heat maps of local packing as previously described (*Figure 4C*). The entire dataset was analyzed computationally in order to test the idea that chain membership depended on MT proximity. For each MT, we calculated the number of neighboring MTs within a given search radius and compared these values for single ('unchained') MTs and those that were part of a chain (*Figure 4D*). This analysis showed that in controls, the single MTs and chained MTs had similar numbers of neighbors, that is, meshed MTs were not more likely to be in tightly packed regions of the fiber. However, in the TACC3 overexpressing fibers, the MTs that were part of a chain had significantly more neighbors than single MTs (*Figure 4D*), this pattern was seen for search radii >50 nm and was not seen if the dataset was randomized (*Figure 4D*, inset, see 'Materials and methods'). The closer proximity of chained MTs vs single MTs indicates that the mesh influences MT positioning, effectively pulling them closer together. An alternative possibility is that the mesh is passive and only encapsulates MTs if they are in close proximity. However, this possibility is less likely because MTs in control fibers exhibit no difference in their proximity relative to their chain status.

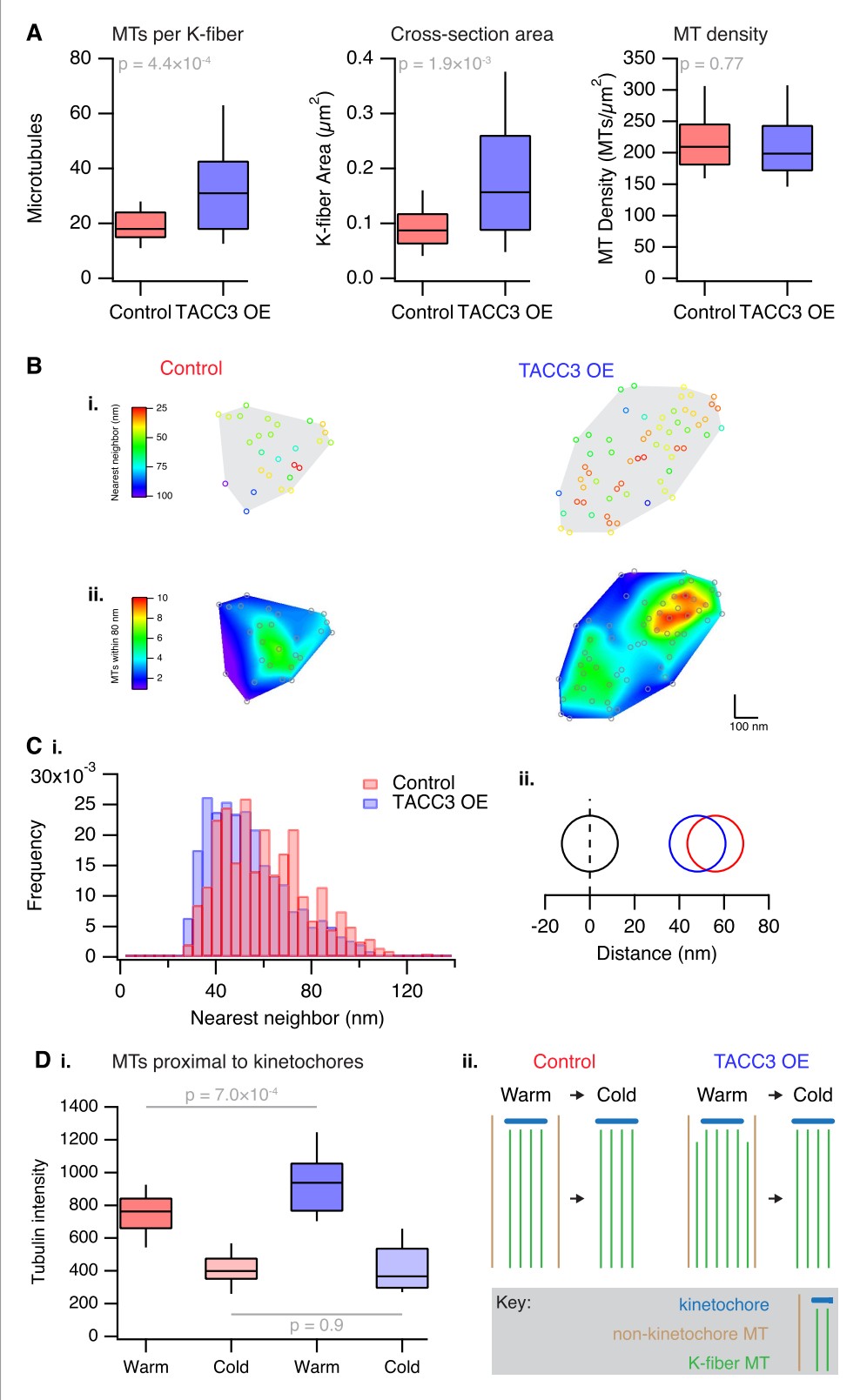

**Figure 3**. Analysis of MT packing within a K-fiber. (**A**) Box plots showing the number of MTs per K-fiber, the cross-sectional K-fiber area, and the density of MTs in the K-fiber. $N_{fiber}$ = 26 (control), 37 (TACC3 OE). Box plots show the median, 75th and 25th percentile, and whiskers show 90th and 10th percentile. p values from Welch's t-test are shown. (**B**) Spatial maps of MTs (circles) in a representative K-fiber from a control (left) or TACC3 overexpressing cell

*Figure 3. continued on next page*

*Figure 3. Continued*

(right) color-coded to show (**i**) the distance to the nearest neighbor or (**ii**) overlaid on a heatmap to show the number of MTs within 105 nm (center-to-center distance), 80 nm (edge-to-edge distance). (**C**) Histogram to show the frequency of distances to the nearest neighboring MT (**i**). A representation of the change in average (median) MT spacing to the nearest neighboring MT that is caused by TACC3 overexpression (**ii**). $N_{MTs} = 500$–1324. (**D**) Analysis of tubulin intensity in the vicinity of kinetochores by confocal microscopy. Box plot to show the distribution of median values per cell of tubulin intensities in a sphere surrounding the kinetochore (**i**). Box plots show the median, 75th and 25th percentile, and whiskers show 90th and 10th percentile. p values from one-way Anova with Tukey's *post hoc* test are shown. $N_{cell} = 20$–25, $N_{kinetochore} = 2823$–3533. Schematic diagram of the changes in K-fibers induced by TACC3 overexpression (**ii**). K-fibers are thicker because MTs that are not stably attached to the kinetochore are recruited to the K-fiber bundle.

The following figure supplement is available for figure 3:

**Figure supplement 1**. Inducible expression of GFP-TACC3 in HeLa cells to alter the composition of mesh.

## Bundling of MTs by mesh components

The two possibilities for mesh stabilization of K-fibers are shown in (*Figure 5A*). To test the possibility that the mesh can influence MT positioning in K-fibers, we turned to an in vitro assay. Fluorescently labeled MTs assembled in vitro and stabilized with taxol were incubated with proteins, and any effect on MT bundling was observed by light or EM. To reconstitute the mesh component containing TACC3, a protein mixture comprising clathrin, TACC3, ch-TOG, and GTSE1 was prepared and phosphorylated by Aurora-A kinase (see 'Materials and methods'). As a positive control, we used PRC1, which is known to bundle MTs (*Mollinari et al., 2002*). As a negative control, we used an equivalent amount of GST and MBP-His$_6$ protein as in the test condition, phosphorylated by Aurora-A kinase. Thick bundles of MTs could be seen by fluorescence microscopy for the TACC3-containing complex and for PRC1 but not for the negative control (*Figure 5B*). These images indicated specific bundling activity for the TACC3 complex.

To look in more detail at the bundled MTs, we analyzed the control, PRC1, and TACC3 complex conditions by EM. In the PRC1 and TACC3 complex conditions, pairs of MTs were interconnected by short electron dense connectors (*Figure 5C*). In control conditions, MTs were randomly oriented, but on occasions when they were in close proximity, no density was seen.

These MT-bundling experiments were complicated by the purification of several large proteins (*Figure 5—figure supplement 1*). As an alternative source, we immunoprecipitated protein complexes, which contained clathrin and TACC3 directly from mitotic spindle of HeLa cells at metaphase (*Figure 5—figure supplement 2A,B*). We observed bundling of Taxol-stabilized MTs in vitro, using this complex specifically (*Figure 5—figure supplement 2C*).

Together these experiments indicate that the TACC3 complex can crosslink MTs and drive the recruitment of stabilized MTs into bundles. They are not compatible with a passive role for the mesh in responding to MT positioning. In the context of the K-fiber, this suggests a role for the mesh in maintaining fiber integrity by tethering MTs together and influencing MT positioning.

## Overexpression of TACC3 influences MT trajectories within K-fibers

To look further for any evidence of a role for the mesh in MT positioning, we examined the trajectories of MTs within each fiber. K-fibers are bundles of parallel MTs, and we sought to characterize how parallel the MTs are and to test if this is altered by TACC3 overexpression. Deviations from parallel would suggest that the mesh tethers MTs closer together and interferes with their trajectory. Qualitatively, we could see that MTs in TACC3 overexpressing K-fibers were more disorganized (*Figure 6A*, *Video 2* and *Video 3*). To measure parallelism more rigorously and to allow comparison of multiple K-fibers, we used several computational approaches (described in 'Materials and methods'), which allowed us to normalize the 3D trajectory of each fiber and then examine the deviation of MT trajectories from this vector. K-fibers in cells overexpressing TACC3 had a higher proportion of MTs that deviate from parallel. They frequently had significantly larger polar angles compared to MTs in K-fibers from control cells (*Figure 6B,C*). The deviation of MTs from a completely

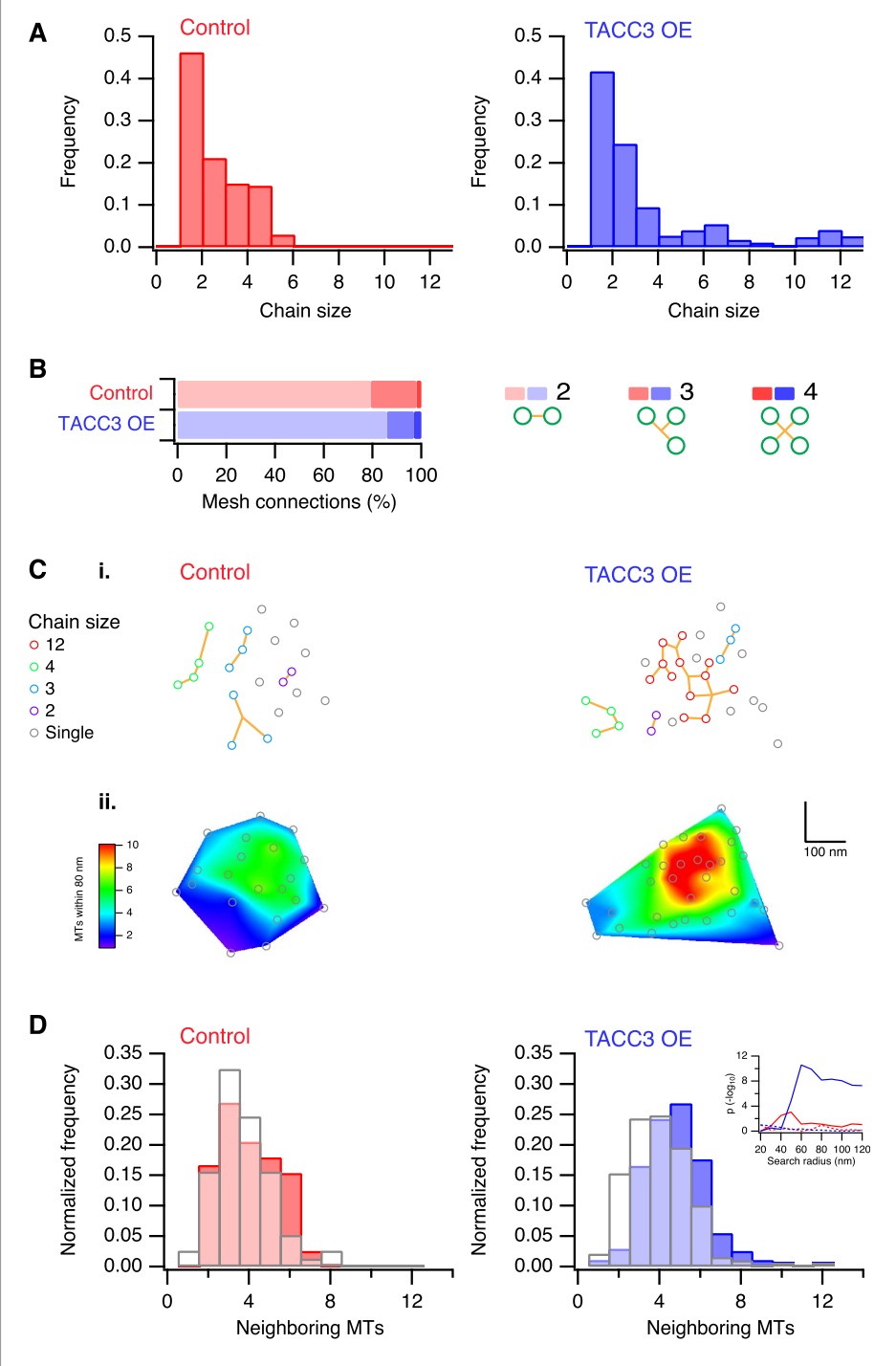

**Figure 4**. Analysis of MTs captured by mesh, their connectivity, and proximity. (**A**) Histogram to show frequency of MT chain sizes detected in single section tomograms. MT chains are collections of MTs that are connected by mesh within a section. Note that single MTs do not feature. (**B**) Bar chart to show the proportion of mesh that is bipolar, tripolar, and tetrapolar, as a percentage of total mesh connections. (**C**) Analysis of MT connectivity and proximity. (**i**) Spatial maps of MTs (circles) in a representative K-fiber from a control (left) or TACC3 overexpressing cell (right). Chains of MTs are shown in color, gray circles show MTs that are not detectably connected to other MTs by mesh. Orange lines represent the mesh connections schematically (see *Figure 4—figure supplement 1*). (**ii**) Spatial maps are shown overlaid on heatmaps for the same fibers to show the number of MTs within 105 nm (center-to-center distance), 80 nm (edge-to-edge distance). Note that single (unchained) MTs are attached to the mesh, but that the MT to which they are attached is outside of the section. Tomogram thickness, 43.6 nm (Control),

*Figure 4. continued on next page*

*Figure 4. Continued*

28.8 nm (TACC3 OE). (**D**) Histogram to show frequency of MTs with a given number of neighboring MTs within 100 nm (center-to-center distance). MTs connected to others by mesh are shown in color, and single MTs are shown in white. Inset shows the p-value for comparisons calculated using search radii from 20 nm to 120 nm (center-to-center distance). Dotted lines show the same analysis following randomization of the chain membership data.

The following figure supplement is available for figure 4:

**Figure supplement 1**. Examples of MTs chains and associated mesh.

parallel condition is easier to visualize, by examining the Cartesian intersection coordinates of each MT vector with an x-y plane at a given distance (z) from a common origin (see 'Materials and methods'). These scatter plots show that MTs in TACC3 overexpressing cells deviate further from parallel than those in control cells (*Figure 6D*). The fraction of MT vectors intersecting this plane at radial distances of <10 nm from the center is 0.39 and 0.18 in control and TACC3 overexpressing cells, respectively. In other words, there are half as many parallel MTs after TACC3 overexpression.

Where are these deviant MTs? If they resided towards the periphery of the fiber, they may represent 'extra MTs', perhaps non-kinetochore MTs, that became enmeshed in the fiber when TACC3 was overexpressed. Alternatively, if the deviant MTs are throughout the fiber, this could be evidence for a role of the mesh in influencing MT spacing and packing within the fiber. To assess this, we plotted the polar and azimuthal angles as a function of distance from the K-fiber center (*Figure 6E*). These plots show that deviant MTs in TACC3 overexpressing cells are distributed at all distances from the center of the fiber. In controls, there was a weak tendency for deviant MTs to be at the fiber periphery, but the overall extent of trajectory deviancy was lower than in TACC3 overexpressing cells (*Figure 6C,E*). The lack of relation between distance from the fiber center and the deviancy from the fiber trajectory of MTs in fibers from TACC3 overexpressing cells indicates that the mesh plays an influential role in organizing MTs *within* the fiber. Together our results show that the change in composition of the mesh, caused by TACC3 overexpression, results in tighter local packing of MTs, more interconnectivity and disruption of the parallel organization of MTs within a K-fiber.

## Mitotic problems associated with TACC3 overexpression

What are the functional consequences of the ultrastructural changes in MT organization *within* K-fibers? To address this question, we used live-cell imaging of our stable inducible cells expressing H2B-mCherry, growing asynchronously and compared mitotic progression in cells, where GFP-TACC3 expression was induced vs not induced. *Figure 7* shows that TACC3 overexpression increases the time taken for cells to congress all chromosomes to the metaphase plate (prometaphase-to-metaphase). In addition, the time from last chromosome alignment to the onset of anaphase (metaphase-to-anaphase) is also longer (*Figure 7*). No change was seen in the duration of anaphase (anaphase-to-telophase). Interestingly, these changes in mitotic progression are similar to those seen following depletion of TACC3 (*Lin et al., 2010*; *Cheeseman et al., 2013*) (*Figure 7—figure supplement 1*). We conclude from these experiments that mitosis is sensitive to the levels of TACC3 and that this sensitivity correlates to changes in MT organization within the K-fiber.

## Discussion

In this paper, we described the mesh: a network of MT connectors in K-fibers. We showed that the mesh is comprised of different connector types and is a major component of the K-fiber. Overexpression of TACC3 alters the mesh and changes the trajectory of MTs within the fiber, indicating that the mesh is a determinant of K-fiber structural integrity. This manipulation causes defects in mitosis and establishes that TACC3 levels at the spindle must be regulated for optimal mitosis.

Since the first EM studies of mitotic spindles, it has been clear that K-fiber MTs are held together throughout their length, presumably to function as an integrated unit (*Hepler et al., 1970*; *Rieder, 1981*; *Witt et al., 1981*). The importance of inter-MT connections along the length of the K-fiber was

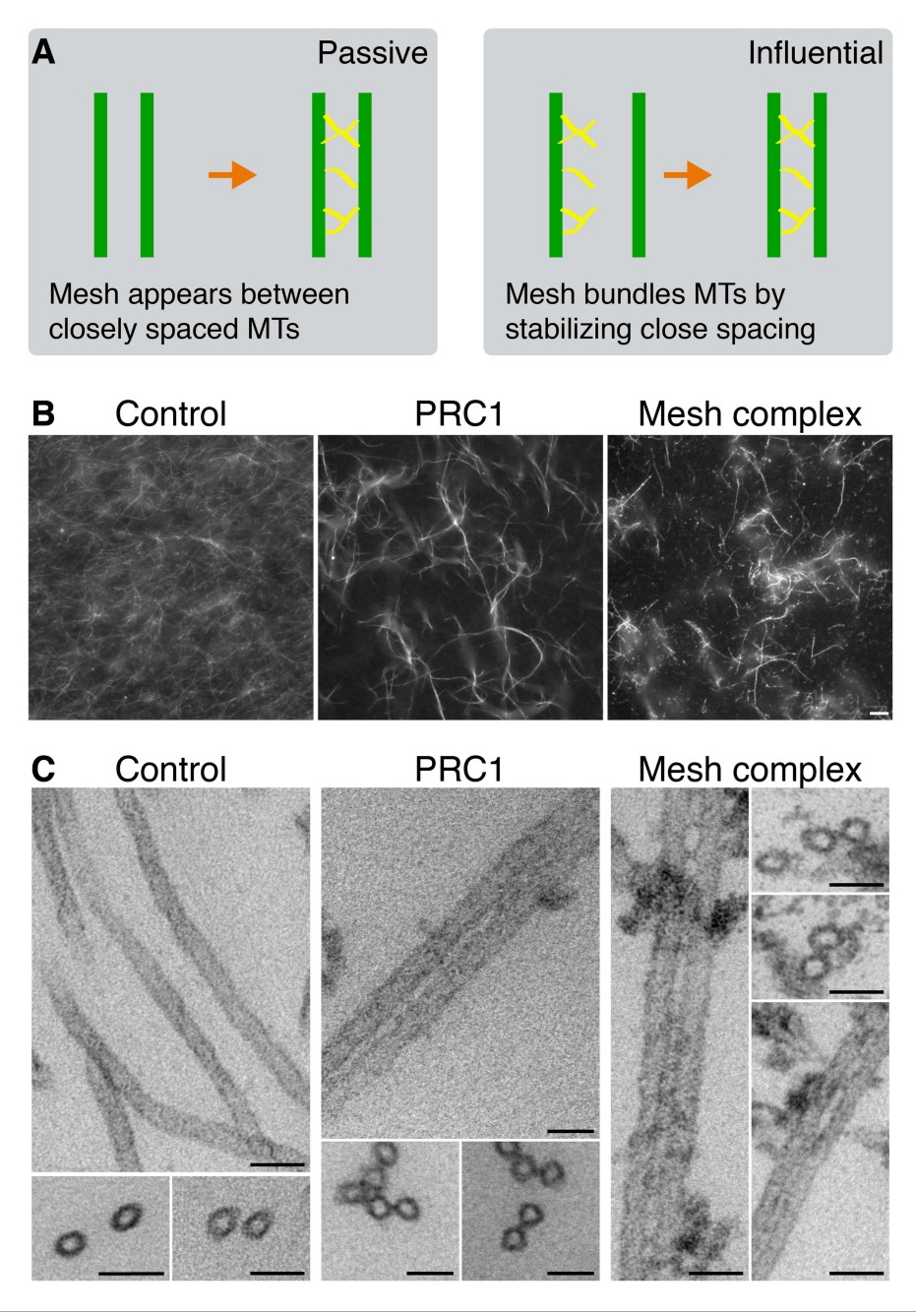

**Figure 5**. Bundling of MTs in vitro by mesh components. (**A**) Cartoon to illustrate two potential models for mesh function. In the passive model (left), MT distances are set by some other factor, and the mesh fills in the gaps to connect MTs. In the influential model (right), the mesh bundles MTs by favoring close spacing, this in turn influences MT spacing. (**B**) Representative fluorescence micrographs of rhodamine-labeled MTs incubated with the indicated proteins. Control (210 nM MBP-His$_6$, 100 nM GST), PRC1 (2 μM His$_6$-PRC1), and Mesh complex (10 nM MBP-ch-TOG-His$_6$, 100 nM clathrin, 100 nM GST-TACC3-His$_6$, 100 nM MBP-GTSE1-His$_6$), all phosphorylated by TPX2(1–43)/Aurora-A. Scale bar, 10 μm. (**C**) Representative electron micrographs of MTs incubated with the indicated proteins as described in **A**. Pelleted material was chemically fixed, embedded in resin, sectioned, and imaged. Scale bar, 50 nm.

The following figure supplements are available for figure 5:

**Figure supplement 1**. MT bundling using purified components.

**Figure supplement 2**. MT bundling using mesh complex immunoisolated from mitotic HeLa cells.

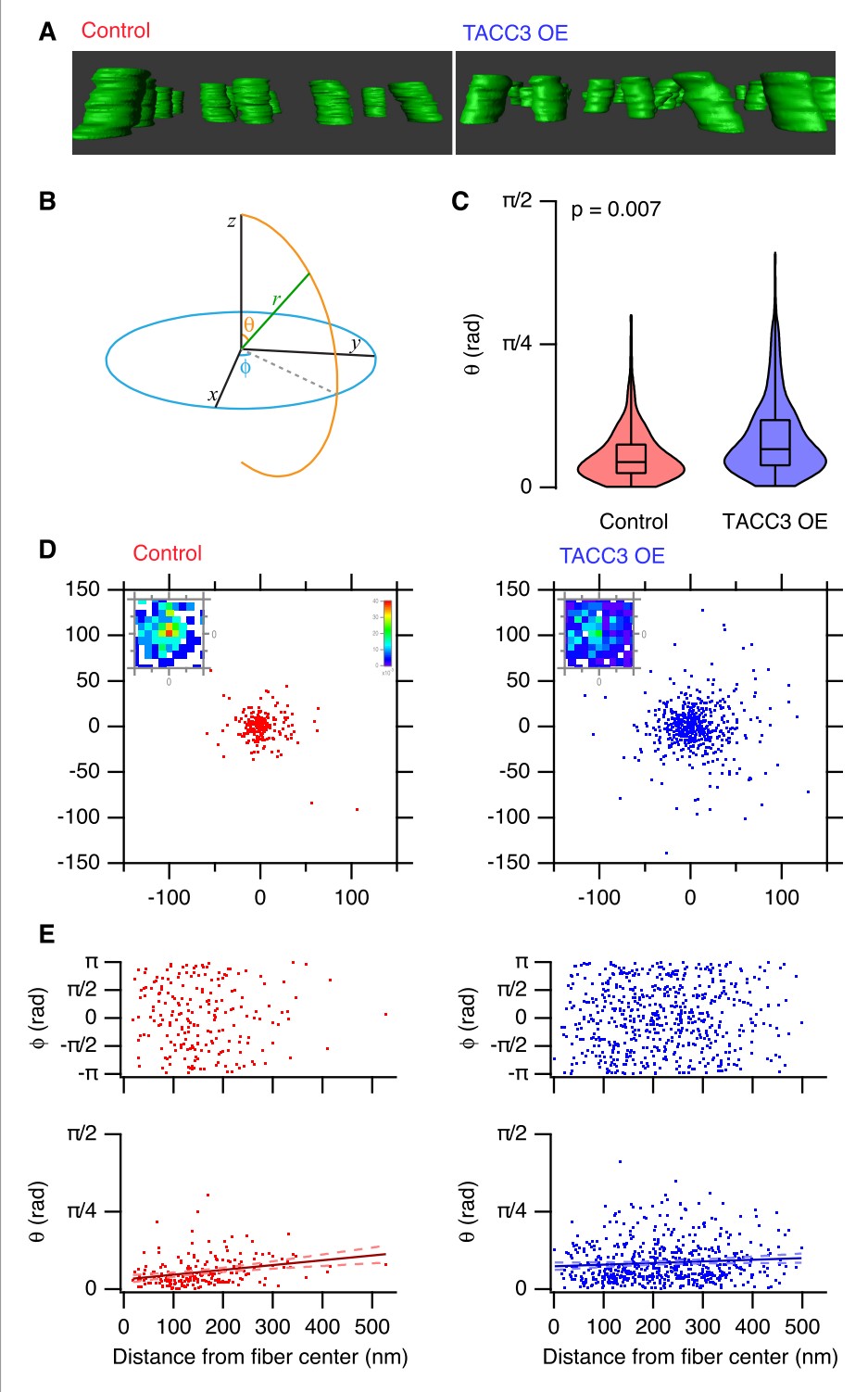

**Figure 6**. Analysis of MT trajectories within a K-fiber. (**A**) Representative 'side views' of rendered MTs in single tomograms of individual K-fibers from control (left, red, *Video 2*) or TACC3 OE (right, blue, *Video 3*) cells. Tomogram thickness, 66.4 nm (Control) trimmed to 34.2 nm, 34.2 nm (TACC3 OE). For scale, MTs are 25 nm in diameter. (**B**) *Aide memoire* of the spherical coordinate system. Trajectories of MTs (green line) in K-fibers were defined and normalized via Euler's rotation (see 'Materials and methods') such that the overall trajectory of all fibers pointed to the zenith. Measurements of polar angle (θ) and azimuthal angle (φ) were made from the normalized sets and are presented in **C**–**E**. (**C**) Violin plots of the polar angles for MTs in K-fibers from control and TACC3
*Figure 6. continued on next page*

*Figure 6. Continued*

overexpressing cells. Box plots show the median, 75th and 25th percentile, and whiskers show the minimum and maximum. Violins show a kernel density estimate of the data and are trimmed to the minima and maxima. Comparison of the median MT angle from each K-fiber, Wilcoxon Rank Sum Test, p = 0.007, $N_{fiber}$ = 12–15. (**D**) Illustration of the deviation of MTs from the fiber axis caused by TACC3 overexpression. Cartesian coordinates of the intersection of individual MT vectors that start from a common origin (0, 0, 0), with an x-y plane at z = 100 nm. To account for the difference in number of MTs, the insets show a normalized 2D histogram of these coordinates cropped to a 40 × 40 nm square centered at 0, 0, 100. (**E**) Plots of azimuthal angle (above) and polar angle (below) as a function of distance from the fiber center (defined as described in the 'Materials and methods'). Line of best fit is shown with 95% confidence bands (dashed line), $r^2$ = 0.07 (control), and 0.007 (TACC3 overexpression).

demonstrated by severing K-fibers at their midpoint (*Spurck et al., 1997*). This resulted in two K-fiber stubs attached to the pole and kinetochore. The stubs do not immediately collapse but instead maintain their structural integrity. Our work identifying the mesh suggests that it is the structure that holds MTs together. This makes sense structurally, but there is a possibility that the mesh is passive and simply connects MTs in the fiber, wherever they may be. However, our work points to a role for the mesh in influencing MT position within the K-fiber. The evidence for this is: first, in K-fibers overexpressing TACC3, interconnected MTs are closer together than those not contacted by mesh. Second, a complex containing TACC3 was able to bundle MTs in vitro. Third, mesh in TACC3 overexpressing fibers pulls MTs out of alignment. In this way, the mesh plays an active role in MT positioning in K-fibers by favoring inter-MT interactions of defined distances. An extension of this is that the mesh can actively pull MTs together, in an energy-consuming process. We have no evidence for this intriguing possibility, which is compatible with our findings.

It seems counterintuitive that overexpression of a mesh component results in skewed MTs; shouldn't it make the MTs in the fiber 'more parallel'? However, it is likely that the mesh is made of other proteins, besides the TACC3–ch-TOG–clathrin complex, for example, CHD4, HURP/DLGAP5, and HSET/KIFC1 (*Mountain et al., 1999*; *Sharp et al., 1999*; *Koffa et al., 2006*; *Yokoyama et al., 2013*). We favor the idea that because we have increased only one component of the mesh experimentally and not others, we do not see uniform changes in spacing and packing. This seems plausible as previous analysis determined TACC3–ch-TOG–clathrin was the shortest crosslinks in K-fibers (*Booth et al., 2011*), and so an imbalance of connector sizes in the mesh is predicted to skew MT trajectories.

The spindle matrix was originally proposed as a microtrabecular lattice that assists the motors and MTs of the mitotic spindle during chromosome movement (*Pickett-Heaps et al., 1982*). The existence

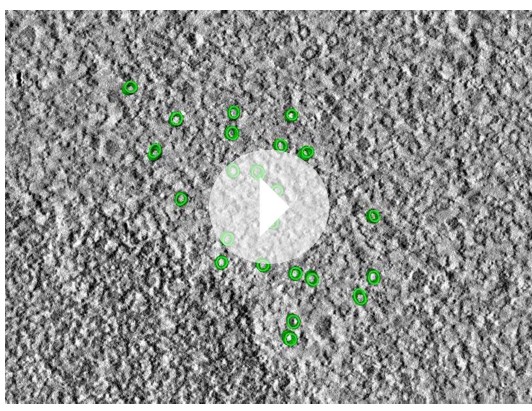

**Video 2.** Example of MT organization in a K-fiber from a control cell. Tomogram of a K-fiber. MTs (green) were rendered by hand. All segmentation was smoothed in Amira.

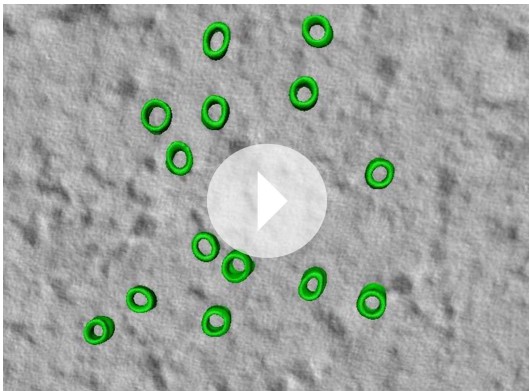

**Video 3.** Example of MT organization in a K-fiber from a GFP-TACC3 expressing cell. Tomogram of a K-fiber. MTs (green) were rendered by hand. Pale green is used to highlight two highly deviant MTs. All segmentation was smoothed in Amira.

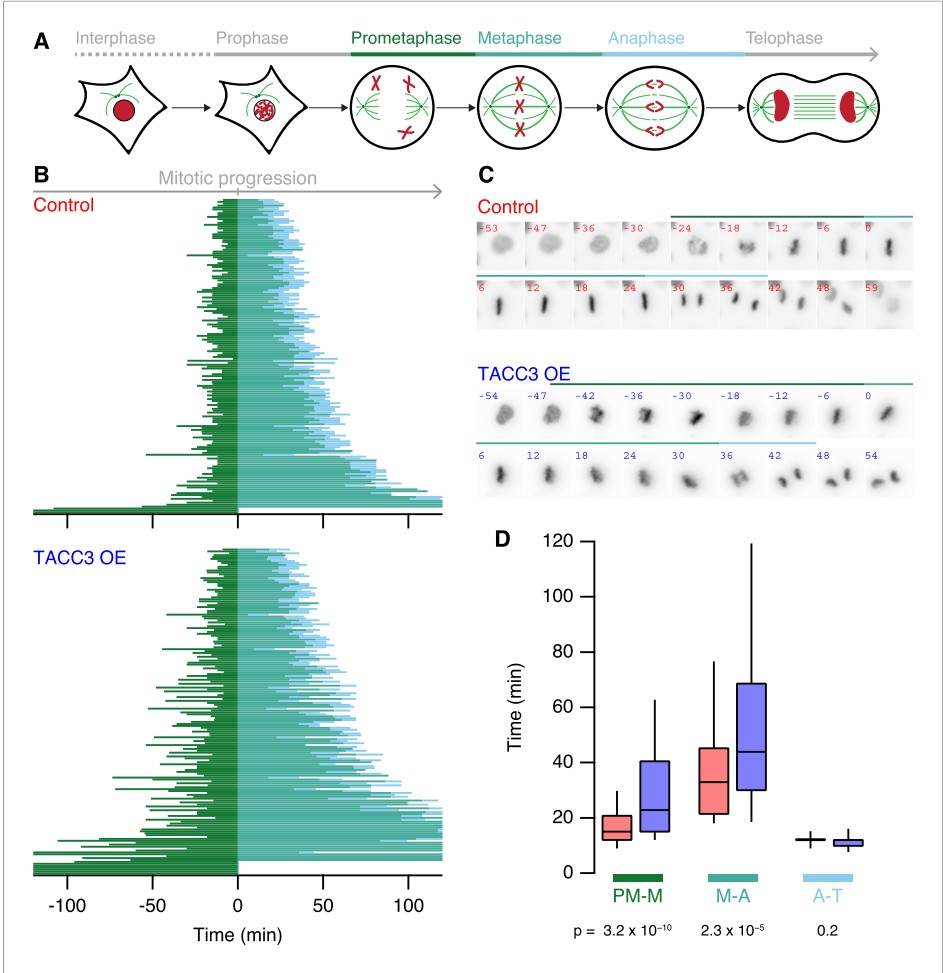

**Figure 7**. Mitotic consequences of TACC3 overexpression. (**A**) Cartoon representation of the stages of mitosis captured in live-cell imaging experiments. Cells expressing H2B-mCherry were staged as indicated with the transition to prometaphase marked by nuclear envelope breakdown; transition to metaphase marked by alignment of the last chromosome to the metaphase plate; to anaphase marked by first sign of chromosome segregation and transition to telophase marked by decondensation of H2B-mCherry. These events allowed us to assign prometaphase-to-metaphase (PM-M, dark green), metaphase-to-anaphase (M-A, light green), and anaphase-to-telophase (A-T, light blue). (**B**) Mitotic stage duration for individual cells. Time taken to go from prometaphase-to-metaphase (PM-M), metaphase-to-anaphase (M-A), or anaphase-to-telophase (A-T) is shown for control (red, above) or GFP-TACC3 overexpressing (blue, below) cells expressing H2B-mCherry. Each line represents a single cell. The time when the last chromosome aligned was set as 0 (metaphase). Cells that did not achieve metaphase halt at 0. Note that the time axis is truncated to show a 4 hr window centered on metaphase. $N_{cell}$ = 163–169, $N_{exp}$ = 4. (**C**) Still images of example cells from live-cell imaging experiments to determine mitotic progression. Detection of H2B-mCherry is shown on an inverted grayscale. Time in min is shown and the colored bars represent the staging, as in **A** and **B**. (**D**) Box plots to summarize mitotic progression experiments. Time taken to go from prometaphase-to-metaphase (PM-M), metaphase-to-anaphase (M-A), or anaphase-to-telophase (A-T) is shown for control (red) or GFP-TACC3 overexpressing (blue) cells. Whiskers show 90th and 10th percentiles.

The following figure supplement is available for figure 7:

**Figure supplement 1**. Mitotic consequences of TACC3 depletion.

of the matrix has been elusive, although several large macromolecular complexes have been proposed as matrix components (*Pickett-Heaps et al., 1984*; *Chang et al., 2004*; *Ma et al., 2009*; *Johansen et al., 2011*; *Schweizer et al., 2014*). Although superficially similar, we do not think that the K-fiber mesh is the ultrastructural correlate of the spindle matrix. The mesh that we have observed

exists between MTs within K-fibers (rather than between K-fibers) and is most likely composed of MT-associated proteins that function on K-fiber MTs. Most models of the spindle matrix envisage an elastic milieu that encompasses the entire spindle, yet is independent of the MTs (*Scholey et al., 2001*; *Yao et al., 2012*). Despite this, micromanipulation experiments have left open the possibility that an anisotropic matrix could exist along K-fibers (*Gatlin et al., 2010*), and this would be compatible with our observations. The K-fiber mesh is in agreement with theoretical and experimental work showing that lateral MT connectors are important for MT arrays to sustain physical forces in cells (*Brangwynne et al., 2006*).

We used overexpression of TACC3 as our primary tool to manipulate the mesh. Progression through mitosis is slower under these conditions due to problems in chromosome congression and satisfying the spindle assembly checkpoint. Interestingly, a similar phenotype is observed when TACC3 is depleted from the spindle (*Lin et al., 2010*; *Booth et al., 2011*; *Cheeseman et al., 2013*). This indicates that mitosis is sensitive to TACC3 levels. We propose that this sensitivity reflects the robustness of the K-fibers. Too little TACC3–ch-TOG–clathrin and the mesh cannot strengthen the fibers adequately. Too much, and the MTs within the fibers become less parallel and oversupported. Previous work has suggested that hyperstabilization of K-fibers is counter-productive, leading to errors in mitosis (*Bakhoum et al., 2009*). TACC3 levels are altered in several human cancers, for example, ovarian, non-small cell lung cancer, and bladder (*Schmidt et al., 2010*), and fusions between TACC3 and FGFR3 are reported in glioblastoma and bladder cancer (*Singh et al., 2012*; *Williams et al., 2013*). Moreover, the amount of TACC3–ch-TOG–clathrin at the spindle is limited by the availability of phosphorylated TACC3 (*LeRoy et al., 2007*; *Hood et al., 2013*), and since Aurora-A kinase is amplified or upregulated in a wide range of cancers (*Nikonova et al., 2013*), it is possible that hyperstabilization of K-fibers occurs in these cells. Under any of these conditions, erroneous mitosis leading to aneuploidy may contribute to cancer initiation or progression. While we favor the idea that normal mitosis requires optimal K-fiber stabilization by the mesh, it must be noted that TACC3 has at least one additional function in mitosis (*Gutiérrez-Caballero et al., 2015*), and future work will seek to delineate the role of TACC3 in the mesh vs at the MT plus-ends.

Identifying and locating proteins within the mesh is the next challenge. Potentially, 3D views of K-fibers in situ will allow us to dock molecular structures of mesh components as has been achieved for other cellular systems (*Lucic et al., 2013*). Methods to unambiguously assign specific electron density to a particular protein will enable us to achieve this, and efforts are currently underway in our lab to do this.

## Materials and methods

### Cell Biology

HeLa cells (HPA/ECACC #93021013) were maintained in Dulbecco's Modified Eagle's Medium (DMEM) plus 10% fetal bovine serum (FBS) and 100 U/ml penicillin/streptomycin, in a humidified incubator at 37°C and 5% $CO_2$.

For TACC3 overexpression, HeLa Tet-On cells (Clontech, Takara, Saint-Germain-en-Laye, France) were used to inducibly express GFP-TACC3. HeLa Tet-On cells were maintained in full DMEM with 300 μg/ml G418, and HeLa Tet-On cells with stably integrated pTRE2hyg-GFP-TACC3(KDP) plasmid were maintained with 300 μg/ml G418 and 200 μg/ml Hygromycin B. Expression of GFP-TACC3 was induced with 0.5 μg/ml doxycycline, 24 hr before analysis (EM analysis or mitotic progression experiments). For TACC3 knockdown, cells were transfected with plasmids to co-express GFP and shRNA using GeneJuice according to the manufacturer's instructions 2 days before imaging.

For immunofluorescence, HeLa cells on coverslips transfected to express GFP or GFP-TACC3 were either kept at 37°C or cold-treated for 5 min then fixed with ice-cold methanol for 10 min. Cells were then blocked (PBS with 5% BSA and 5% goat serum) before immunostaining with rabbit anti-alpha-tubulin (Thermo, PA5-19489, 1:500) and mouse anti-CENPA (Abcam, ab13939, 1:500) and Alexa568/Alexa633-conjugated secondary antibodies.

For EM analysis, cells were synchronized using 2 mM thymidine block for 16 hr, release for 6 hr, followed by 9 μM RO-3306 for 16 hr (see *Figure 3—figure supplement 1*). After 30–40 min release, mitotic cells were collected by shake-off and centrifugation at 300×*g* for 2 min at 37°C and resuspended in DMEM containing 20% FBS. Cell synchronization was necessary to harvest sufficient numbers of cells for our analysis. Similar results were obtained with cells synchronized using

nocodazole, and with no synchronization, using a correlative light-EM approach, where a single cell was targeted for freezing.

## Molecular biology

For inducible expression, GFP-TACC3KDP (*Booth et al., 2011*) was inserted into pTRE2hyg vector at *Nhe*I and *Not*I sites. Plasmids for expression of GST-TACC3-His$_6$ and GST-TPX2(1–43) were available from previous work (*Hood et al., 2013*). To make MBP-ch-TOG-His$_6$, the coding sequence of full-length human ch-TOG was amplified by PCR to add *Bam*HI and *Eag*I sites before subcloning into pMALPreHis vector. For MBP-GTSE1-His$_6$, the coding sequence of human G-2 and S-phase expressed 1 was amplified from an IMAGE clone (4138532) to add *Eco*RI and *Bam*HI sites. For His$_6$-PRC1, the coding sequence of human Protein Regulator of Cytokinesis 1 was amplified from an IMAGE clone (2958690) and inserted into pRSETB at *Nhe*I and *Eco*RI sites. Plasmids to co-express GFP with shRNA targeting GL2 or TACC3 (pBrain-GFP-shGL2 and pBrain-GFP-shTACC3) were available from previous work (*Booth et al., 2011*; *Lopez-Murcia et al., 2014*).

## Protein biochemistry

Clathrin was purified from rat liver as described previously (*Hood et al., 2013*; *Lopez-Murcia et al., 2014*). Recombinant human Aurora-A kinase was purchased from EMD Millipore (Watford, UK). Reagents for in vitro MT assembly were from Cytoskeleton Inc (Bioquote, York, UK). All proteins were purified from BL21pLysS bacteria as described previously (*Hood et al., 2013*).

For in vitro MT bundling experiments, the following conditions were used: Control (210 nM MBP-His$_6$, 100 nM GST), PRC1 (2 μM His$_6$-PRC1), and Mesh Complex (10 nM MBP-ch-TOG-His$_6$, 100 nM clathrin, 100 nM GST-TACC3-His$_6$, 100 nM MBP-GTSE1-His$_6$), and visualized by fluorescence microscopy and EM. Concentrations refer to the final concentration with MTs. Proteins were incubated at 30°C for 90 min to allow for phosphorylation by TPX2(1–43)/Aurora-A, 2 μg/ml each. MTs were polymerized from 100 μM tubulin (1:10 labeled to unlabeled) in general tubulin buffer (80 mM K-Pipes, 2 mM MgCl$_2$, 0.5 mM EGTA, pH 7.0) with 1 mM GTP and 6% glycerol for 15 min at 37°C. Diluted 1:20 in general tubulin buffer supplemented with 10 μM taxol (paclitaxel, Sigma) and 1 mM GTP, incubated for a further 5 min at 37°C. To 92.4 μl of protein mixture, 69.9 μl of MTs were added and incubated at RT for 10 min. Then 7.25 μl of this was fixed with glutaraldehyde in BRB80 (1.45 μl; 0.1% final), then diluted with 10 μl 70% glycerol/BRB80. 6 μl was spotted onto glass slides, overlaid with a cover slip, and sealed with nail varnish. The remainder was split into two and spun for 90 min RT 21,000×*g* and the pellets fixed in 1% glutaraldehyde for 2 hr RT, followed by post-fixing with 3% glutaraldehyde, 15 min RT. Following three washes in PBS, samples were osmicated in 1% osmium tetroxide, 1 hr RT, washed in PBS (3 × 20 min), then dH$_2$O (2 × 20 min), and into 30% ethanol for 20 min. Samples were incubated in 0.5% uranyl acetate in 30% ethanol for 40 min. Samples were dehydrated and infused with resin and cured at 60°C for 2 days. Sections were cut and visualized by transmission EM. The MT bundling assay by fluorescence microscopy alone was carried out more than five times. The experiment where the same samples were visualized by fluorescence microscopy and EM was performed twice using different protein preparations.

Immunopurification of the mesh complex was as described previously (*Booth et al., 2011*). Beads from this procedure were used directly for MT-bundling experiments as described above.

## Light microscopy

For the mitotic progression assays, $6.8 \times 10^4$ cells/ml were seeded in 12 well plates and transfected with H2B-mCherry. GFP-TACC3 expression was induced the day after transfection. 24 hr later, the cells were imaged on a Nikon Ti epifluorescence microscope with 40× ELWD objective (0.6 NA) and a Coolsnap Myo camera (Photometrics (Tucson, AZ)) for 14 hr using NIS Elements AR software; H2B-mCherry was imaged once every 3 min, and GFP-TACC3 monitored once every 30 min. Cells were kept at 37°C, in supplemented CO$_2$-independent medium containing doxycycline (0.5 μg/ml) for cells with GFP-TACC3 induction. Light intensity and exposure were minimized to avoid light-induced cell damage. Analysis of mitotic staging was by manual inspection of videos, with automated time look-up. The same microscope was used to assay the bundling of fluorescent MTs.

For analysis of MTs in the proximity of kinetochores, a similar method to that previously described was used (*Cheeseman et al., 2013*). Briefly, confocal imaging was performed on a spinning disc

confocal system (Ultraview Vox, Perkin Elmer) using a 100× ~1.4 NA oil immersion objective lens with a Hamamatsu C10600-10B ORCA-R2 camera. Fixed cells expressing GFP or GFP-TACC3, stained as described above, were excited at 488 nm, 561 nm, 405 nm, and 640 nm, and z-sections were taken every 0.25 μm. OME-TIFF image stacks were transferred to IMARIS (Bitplane) and subjected to spot detection in the CREST/CENP-A channel with a minimal spot size of 0.58 μm to detect kinetochores. Spots that were not properly assigned (~5%) were corrected manually. The mean pixel densities in this sphere were background subtracted, and the median intensity per cell was found using IgorPro.

## Electron microscopy

HPF was performed using either EM PACT2 or EM HPM100 (Leica Microsystems, Milton Keynes, UK). Mitotic cells were transferred to 100 μm-depth membrane carriers or Type A 100 μm-depth carriers for the EM PACT2 or EM HPM100, respectively. Once frozen, carriers were transferred in liquid nitrogen to the EM AFS2 (Leica Microsystems) FS machine, which was precooled to −90°C. Carriers were immersed in a FS medium of 1% osmium tetroxide, 0.5% uranyl acetate, and 5% $H_2O$ in acetone. Cells were brought to RT over 67 hr: 27 hr at −90°C; ramp to −60°C over 15 hr; 8 hr at −60°C; ramp to −30°C over 15 hr; 1 hr at −30°C; ramp to 4°C over 1 hr; cells were taken to the fume hood to reach RT. Cells were embedded in epoxy resin (TAAB) and polymerized at 60°C for 48 hr. Sections (70 nm) were cut using an EM UC6 (Leica Microsystems) and post-stained using uranyl acetate and Reynold's lead citrate. Sections were imaged using the FEI Tecnai G2 Spirit BioTWIN microscope at 100 kV. Putative K-fibers were identified as bundles of more than 10 MTs in sections taken orthogonal to the spindle axis. K-fibers were imaged 1 μm away from the kinetochore, so as not to include any kinetochore-MT linkages (*Dong et al., 2007*). Each tilt series was taken using TIA software (TEM imaging and Analysis, FEI) from +50° to −50° in 1° steps. The IMOD etomo package was used to generate tomograms from these tilt series (*Mastronarde, 1997*). Tomogram thickness ranged from 28.8 to 66.4 nm.

## Data analysis

For visualization, 3D rendering of tomograms was done in Amira 5.6.0 (Visualization Sciences Group, FEI, Eindhoven, Netherlands) using a combination of manual and automated detection. Each MT was rendered individually by hand, and then any 3D density connecting the MTs was automatically detected throughout the volume of the tomogram, using the average gray value of the MTs as an initial threshold for segmentation. The space between MTs in a K-fiber is less granular than the surrounding cytoplasm, which improves mesh detection. This unbiased method superseded our initial attempts at hand-rendering the mesh and allowed for unbiased quantification and visualization of the mesh. Note that measurements are taken from the unsmoothed segmentation maps. The segmentation method detects unbroken density connected to MTs. However, after smoothing for visualization, breaks in the mesh appear. In Amira, MTs were smoothed using unconstrained smoothing, and the mesh was rendered with smooth surface (50 iterations, 0.6 lambda). Hand-rendered mesh in *Figure 1B* was smoothed with constrained smoothing and then smooth surface generated (20 iterations, 0.6 lambda).

In longitudinal sections, K-fibers were defined as bundles of MTs contacting both the kinetochore and the pole (*McDonald et al., 1992*; *Booth et al., 2013*). In orthogonal sections, K-fiber bundles were defined as collections of 10 or more MTs, using an 80 nm boundary around each MT (105 nm from the MT center). This distance corresponds to the longest inter-MT bridges visualized previously by 2D EM (*Booth et al., 2011*).

For analysis of MT packing, coordinates of each MT at the midpoint of the tomogram were collected, and the area of a convex hull that enclosed the coordinates was found. The distance to the nearest neighboring MT and the number of MTs within 80 nm (105 nm from the MT center) were calculated. Heat maps were generated by Voronoi interpolation of 3D coordinate sets that comprised the x-y coordinates together with the number of MTs within 80 nm in the z dimension.

To analyze the mesh connections and MT chaining, individual connections in rendered tomograms were classified and recorded together with the chain sizes for each K-fiber. Coordinate sets were supplemented with chain membership data. This was used for an automated comparison of the number of neighboring MTs within a given search radius, comparing chained vs single MTs. The analysis was performed from 20 to 120 nm, and non-parametric Wilcoxon–Mann–Whitney two-sample rank test used to test the hypothesis that there was no difference in the number of neighboring MTs. Chain membership data were then randomized and the analysis repeated. Note that chain membership is

probabilistic and is a function of tomogram thickness and mesh density/MT interconnectivity. 'Single MTs' are those that are not detected to be attached to another but are likely to be attached to other MTs in a subsequent slab of K-fiber.

For analysis of MT trajectories, 3D coordinates for each MT at the bottom and top of the tomogram were logged in ImageJ and fed into IgorPro. Custom-written procedures ran through the following steps. First, coordinates were compiled into individual MT matrices. Second, the center of the bundle was found using farthest-point clustering, and then the distance of each MT from this point was calculated. Third, the fiber direction was 'normalized', that is, rotated through 3D space so that most MTs pointed towards the zenith. This was necessary because of the variability in the axes of different K-fibers relative to sectioning and the resultant tomograms. To do normalization, all MTs were multiplied by rotation matrices

$$R = R_z(\alpha)R_y(\beta)R_x(\gamma),$$

for (in radians) $0 \leq \alpha \leq 2\pi$, $0 \leq \beta \leq \pi/2$, $\gamma = 0$ in $180/\pi$ increments. The Cartesian distance of the orthogonal projection of each MT on the reference plane after rotation was summed, to find the rotation that produced the minimum value. The spherical coordinates, given by

$$r = \sqrt{x^2 + y^2 + z^2},$$

$$\theta = \cos^{-1}\left(\frac{z}{\sqrt{x^2 + y^2 + z^2}}\right),$$

$$\varphi = \tan^{-1}\left(\frac{y}{x}\right),$$

of the MTs set at optimal rotation, were used for plotting. Fourth, a 2D plot to visualize the degree of MT deviancy was generated. To do this, the point at which each 3D MT vector starting at the origin intersects an x-y plane that was set arbitrarily at z = 100 nm was calculated and plotted. Note that the even radial dispersal in *Figure 6D* and the even spread of azimuthal angles in *Figure 6E* show that capture of tilt series was randomized, and the data set has no bias towards a particular trajectory. Several other strategies were explored to analyze deviations in trajectory vs the fiber axis. These were: examining the variance in trajectory angles, pairwise comparison of all MTs in the bundle, and comparison to a reference MT that represented the fiber axis, using spherical rotation and rotating by an average value. These produced similar results, however, the one described here was the most robust and represents our best method for this kind of spatial statistical analysis. The computer code used in the main analysis pipeline can be found as a *Source code 1*.

Images were cropped in Photoshop, and figures were assembled in Illustrator CS5.1. IgorPro 6.36 (Wavemetrics, Portland, OR) was used for all analysis and plotting.

## Acknowledgements

We thank Tom Honnor and Julia Brettschneider, members of the Royle lab and Anne Straube for critical discussion. We also thank Alison Beckett, Sam Williams, and Liam Cheeseman for technical help and Rob Cross for commenting on the manuscript. This work was supported by a project grant from North West Cancer Research (CR928) and by a Senior Cancer Research Fellowship from Cancer Research UK (C25425/A15182) to SJR.

## Additional information

### Funding

| Funder | Grant reference | Author |
| --- | --- | --- |
| Cancer Research UK | C25425/A15182 | Cristina Gutiérrez-Caballero, Stephen J Royle |
| North West Cancer Research Fund | CR928 | Faye M Nixon, Stephen J Royle |

The funders had no role in study design, data collection and interpretation, or the decision to submit the work for publication.

## Author contributions

FMN, Performed all 3D-EM analysis, tomography and rendering, Acquisition of data, Analysis and interpretation of data, Drafting or revising the article; CG-C, Performed the mitotic progression experiments, Acquisition of data, Analysis and interpretation of data; FEH, Performed the in vitro MT experiments, Acquisition of data; DGB, Made the initial observation of 'the mesh' and made cell lines that were essential for the study, Acquisition of data, Contributed unpublished essential data or reagents; IAP, Co-supervised electron microscopy work, Analysis and interpretation of data; SJR, Designed the study, wrote computer code for data analysis, assembled the figures and wrote the paper, Conception and design, Analysis and interpretation of data, Drafting or revising the article

## Author ORCIDs

Stephen J Royle, [iD] http://orcid.org/0000-0001-8927-6967

# Additional files

**Supplementary file**

• Source code 1. Custom written code in IgorPro 6.36 (Wavemetrics) was used for all analysis and plotting.

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
