## [Decision Letter]

Thank you for sending your work entitled “The mesh is a network of microtubule contacts that stabilizes individual kinetochore fibers of the mitotic spindle” for consideration at *eLife*. Your article has been favorably evaluated by Vivek Malhotra (Senior editor), a Reviewing editor, and three reviewers.

The following individuals responsible for the peer review of your submission have agreed to reveal their identity: J Richard McIntosh and Helder Maiato. A further reviewer remains anonymous.

The Reviewing editor and the reviewers discussed their comments before we reached this decision, and the Reviewing editor has assembled the following comments to help you prepare a revised submission.

This is an interesting and one can even say “courageous” paper. Most cell biologists like to work on defined proteins or periodic structures that allow experimental rigor by established criteria. This paper describes work on a component of the mitotic spindle that certainly exists and some of whose components are known, but whose structure is ill-defined and therefore very hard to study. This group has been pursuing aspects of the inter-microtubule “mesh” seen in kinetochore fibers for some time, and this paper takes that work a big step along. However, before the paper can be published, some additional analyses and clarifications are required.

1) While the 3D rendering of the mesh suggests interaction with up to 4 microtubules, careful inspection of the electron densities in the tomograms corresponding to bridges involving more than 2 microtubules is not very convincing from the images provided. For instance, in Figure 1, quadrupolar and tripolar bridges are often interrupted, but are shown as continuous electron densities in the 3D rendered images. Because the criterion of uninterrupted density was chosen for this classification, the authors need to explain why this is not visible in the tripolar and quadrupolar connections and clarify if any threshold was used for 3D-rendering.

2) The authors should show how the mesh compares to an area without microtubules in the spindle and, ideally, to non-kinetochore microtubules. This is important to understand whether the features revealed are specific for K-fibers, as they claim.

3) The authors should characterize K-fiber stability (even if by just a crude cold assay) after TACC3 overexpression. If possible, it would also be nice to compare the results with TACC3 depletion.

4) The decision of the authors to use populations of cells synchronized by thymidine and then RO-3306 (both for 16 hrs) in the analysis of control K-fibers is somewhat puzzling. 16 hrs of late G2 arrest is not benign. For example, centrosome organization becomes abnormal. This in turn affects the organization of the mitotic spindle. Whether the fine structure of K-fibers is affected is not known but there is a danger that it is. The procedure used and its justification should be clearly explained in the main text of the paper.

5) The in vitro reconstitution experiments are performed with a rather ill-defined mixture of four components. The authors should at least illustrate the quality of the individual proteins by Coomassie-stained gels. They should also explain in the main text why they think that all four components need to be added to observe microtubule bundling, whether the HIS-tags present on the individual proteins can contribute to microtubule bundling and also explain the rationale of adding ch-TOG in sub-stoichiometric amounts compared to the other components. Do the authors have any indications that the four components purified by them indeed form a (defined) complex in vitro?

Minor concerns:

1) We couldn't find information about the mitotic stage in which the EM analysis was performed and how K-fibers were identified.

2) In the Introduction, the authors mention that: “chromosome movements are governed by the kinetochore fibers (K-fibers) of the mitotic spindle”. Not all chromosome movements are governed by K-fibers. This is the case for example of chromosome congression mediated by motor proteins, which actually do not need K-fibers to move chromosomes to the spindle equator (work by the Walczak and Khodjakov labs). This should be corrected.

3) In the section “Ultrastructural morphology of the mesh”, the authors state: “The density that we observe as mesh is likely made from many different proteins and the lack of scaling in these measurements suggest that connectors are not multimeric assemblies of a single subunit type”. This sentence is highly speculative and does not really add anything. The lack of scaling could simply reflect conformational changes in highly flexible structures that form inter-microtubule bridges.

4) The analogy with the spindle matrix is unavoidable and the authors disclose this in a nice way in the Discussion, but it does raise the question of whether the mesh is resistant to microtubule depolymerization (related to point 2 in the major concerns, above).

5) The argument against the preferential binding to the microtubule seam is totally dispensable, since one microtubule is able to bind more than 2 connectors.

6) How much is TACC3 being overexpressed and what is the effect over the other proteins of the complex? This important control is missing.

---

## [Author Response]

*1) While the 3D rendering of the mesh suggests interaction with up to 4 microtubules, careful inspection of the electron densities in the tomograms corresponding to bridges involving more than 2 microtubules is not very convincing from the images provided. For instance, in*
Figure 1*, quadrupolar and tripolar bridges are often interrupted, but are shown as continuous electron densities in the 3D rendered images. Because the criterion of uninterrupted density was chosen for this classification, the authors need to explain why this is not visible in the tripolar and quadrupolar connections and clarify if any threshold was used for 3D-rendering*.

The segmentation of K-fiber mesh works by connecting uninterrupted density that touches MTs, in 3D. When we show a single orthoslice (one x-y slice in a tomogram) it is not possible to see the density that exists above and below. In this case the connector appears broken. An additional issue is that when the segmented connector is very thin, the smoothing that we use to generate the 3D-rendering results in breaks in the connector. In the assessment of connector frequency (Figure 4), we were conservative in our classification. We carefully checked each connector individually and only where we could see clear uninterrupted density connecting 2-4 MTs, did we assign the connector.

*2) The authors should show how the mesh compares to an area without microtubules in the spindle and, ideally, to non-kinetochore microtubules. This is important to understand whether the features revealed are specific for K-fibers, as they claim*.

We have added a new figure (Figure 1—figure supplement 2) to address the question of specificity of the mesh for K-fibers. K-fibers are typically in low noise areas of the cytoplasm and this simplifies the detection of mesh associated with K-fiber MTs. Our segmentation method works by detecting density that is connected to MTs. If we break this rule and simply detect in a non-K-fiber region, any density using the same threshold, we pick up particulate material which looks different to the mesh (“non-mesh”, brown, Figure 1—figure supplement 2). Another test we employed early on to give us confidence in mesh detection was to move the segmented MTs to a non-K-fiber area and do the mesh detection. When we do this, we again pick up the particulate density and this amorphous material runs through the lumens of the MTs (Figure 1—figure supplement 2). These images support the idea that the mesh is genuine and associated with MTs, although they do not deal with the specificity for K-fibers versus non-kinetochore MTs. Due to the magnification used for imaging, we do not have many tomograms, which feature MTs where we can be certain that they are not part of a K-fiber. Our feeling is that there is density between non-K-fiber MTs but that it is less abundant, since there are fewer MTs running together—2-6 MTs interpolar MTs in PtK1 cells (25)—and therefore less interconnectivity of MTs.

*3) The authors should characterize K-fiber stability (even if by just a crude cold assay) after TACC3 overexpression. If possible, it would also be nice to compare the results with TACC3 depletion*.

We have carried out a cold-stable assay to assess stable attachment of MTs in cells expressing GFP or GFP-TACC3. These experiments confirmed the “more MTs per K-fiber” as seen by EM. They also showed that after cold treatment there was a similar amount of tubulin in the vicinity of the kinetochores. This was a very useful piece of information (now added in Figure 3). It suggests that the additional MTs seen in K-fibers overexpressing GFP-TACC3 are “passengers” and are not stably attached to the kinetochore. The assay did not indicate that K-fibers are more stable under conditions of TACC3 over-expression, although to really get at this we would need to challenge the K-fibers with depolymerization (using a range of conditions) at different timepoints during spindle assembly. We have previously shown that clathrin depletion, which also removes the complex from spindles, results in K-fibers that are less stable under cold-treatment conditions (35). We also think that further molecular characterization is needed to test what extent the mesh contributes to K-fiber stability.

*4) The decision of the authors to use populations of cells synchronized by thymidine and then RO-3306 (both for 16 hrs) in the analysis of control K-fibers is somewhat puzzling. 16 hrs of late G2 arrest is not benign. For example, centrosome organization becomes abnormal. This in turn affects the organization of the mitotic spindle. Whether the fine structure of K-fibers is affected is not known but there is a danger that it is. The procedure used and its justification should be clearly explained in the main text of the paper*.

For our EM analysis, enrichment of mitotic cells was necessary because we were freezing and sectioning blind to the state of the cells. The controlled release and shake-off allowed us to capture a reasonably pure population of cells at metaphase. It is true that very long arrest with RO3306 (40 h) is detrimental (Loncarek et al., 2010). Shorter arrests like ours (16 h) are not without problems, resulting in 16/44 cells with bipolar spindles in one study (Prosser et al., 2012). It is not known whether multipolar spindles have any difference in the fine structure of the K-fibers compared to bipolar spindles.

To address this point, we have analyzed K-fibers from cells prepared in two different ways and the results suggest that the RO3306 arrest is valid. In the first, we synchronized HeLa cells using nocodazole arrest (16 h) and release; the K-fibers under these conditions are indistinguishable from those in the present study. Secondly, we have begun correlative light-electron microscopy analysis using HPF/FS and tomography of unsynchronized cells. Although we haven’t repeated every aspect of the current study, the K-fibers and mesh are again similar to those reported here.

*5) The in vitro reconstitution experiments are performed with a rather ill-defined mixture of four components. The authors should at least illustrate the quality of the individual proteins by Coomassie-stained gels. They should also explain in the main text why they think that all four components need to be added to observe microtubule bundling, whether the HIS-tags present on the individual proteins can contribute to microtubule bundling and also explain the rationale of adding ch-TOG in sub-stoichiometric amounts compared to the other components. Do the authors have any indications that the four components purified by them indeed form a (defined) complex in vitro*?

To address these comments we have added two new figure supplements to Figure 5. The first figure supplement highlights the limitations in the approach used in the manuscript, while the second describes an alternative way to demonstrate in vitro bundling of MTs by our complex.

*How specific is the bundling*?

First, incubation of MTs with ch-TOG at higher concentrations (∼50 nM) caused bundling of MTs (Figure 5—figure supplement 1). This is why we limited the concentration of ch-TOG to 10 nM in Figure 5. Second, GTSE1 sometimes caused bundling at higher concentrations (Figure 5—figure supplement 1). In an experiment where 10 nM GTSE1 and 10 nM ch-TOG caused no bundling, we could see clear bundling with the addition of clathrin and TACC3, with concentrations of GTSE1 down to 1 nM (Figure 5—figure supplement 1). We do not think that the His tags can contribute to bundling at the concentrations used, since our controls have equivalent amount of His. However, we cannot rule out that His in combination with MAPs causes bundling whereas His in combination with MBP and GST does not.

A further limitation is that the purity of our recombinant proteins is not ideal. The quality of the protein preps used for these experiments is now shown in Figure 5—figure supplement 1 as requested. We also did not assess whether the purified components form a complex for the bundling experiments.

*An alternative method*:

The key issue here is whether the spindle complex that contains TACC3 and clathrin is able to bundle MTs in vitro. To answer this, we purified the complex from mitotic spindle fractions using the immunoisolation method that we used to originally identify the complex (4). Briefly, clathrin was immunoprecipitated from mitotic cytosol (F1) or spindle fractions (F5-7). TACC3 was only associated with clathrin in the spindle fractions (Figure 5—figure supplement 2). These complexes were used for MT bundling experiments. We could see bundling of MTs by clathrin complexes from spindle fraction (F5-7) but not from cytosol (F1). Control anti-myc beads from F1 or F5-7 did not cause bundling (Figure 5—figure supplement 2). Since this approach uses untagged proteins, presumably in their native complex, we think this is strong evidence that the complex can bundle MTs in vitro.

Finally, in further support of the idea that this complex can bundle MTs, we have previously shown (modest) bundling activity when YFP-CHC(1-574) and TACC3(519-838) fragments are phosphorylated by Aurora A kinase and incubated with fluorescent MTs (14).

Minor concerns:

*1) We couldn't find information about the mitotic stage in which the EM analysis was performed and how K-fibers were identified*.

This is now clarified in the Methods section. We released cells from RO3306 for 30-40 min at 37°C. After shake-off and pelleting this gave mainly metaphase cells in either the control or the TACC3-overexpressed condition. K-fibers were identified as bundles of >10 MTs in sections taken approximately orthogonal to the spindle axis. This is now stated under Electron Microscopy. The Data Analysis part of the Methods section explains how K-fibers were identified.

*2) In the Introduction, the authors mention that: “chromosome movements are governed by the kinetochore fibers (K-fibers) of the mitotic spindle”. Not all chromosome movements are governed by K-fibers. This is the case for example of chromosome congression mediated by motor proteins, which actually do not need K-fibers to move chromosomes to the spindle equator (work by the Walczak and Khodjakov labs). This should be corrected*.

We have changed this sentence to read: “Many of the chromosome movements in mitosis are governed by the kinetochore fibers (K-fibers) of the spindle apparatus.”

*3) In the section “Ultrastructural morphology of the mesh”, the authors state: “The density that we observe as mesh is likely made from many different proteins and the lack of scaling in these measurements suggest that connectors are not multimeric assemblies of a single subunit type”. This sentence is highly speculative and does not really add anything. The lack of scaling could simply reflect conformational changes in highly flexible structures that form inter-microtubule bridges*.

We have deleted this sentence.

*4) The analogy with the spindle matrix is unavoidable and the authors disclose this in a nice way in the Discussion, but it does raise the question of whether the mesh is resistant to microtubule depolymerization (related to point 2 in the major concerns, above)*.

We have carried out MT depolymerization experiments using nocodazole (Figure 8). By light microscopy, the signals for GFP-tagged TACC3, clathrin and ch-TOG disappear concomitantly with the mCherry-tubulin signal. We see no evidence for this complex maintaining a matrix-like structure in the absence of MTs.

Author response image 1.TACC3 mesh complex does not persist in a “spindle matrix” after depolymerization of MTs.Still images from live-cell imaging experiments to visualize metaphase HeLa cells expressing mCherry-tubulin and the indicated GFP-tagged protein. Cells were exposed to DMSO (control, left) or nocodazole (10 µM, right) after 1 min. MT depolymerization, monitored by mCherry-tubulin, caused mislocalization of clathrin and ch-TOG. When over-expressed, GFP-TACC3 forms aggregates in the absence of MTs.**DOI:**
http://dx.doi.org/10.7554/eLife.07635.023

In addition to this experiment, we also found using FRAP that TACC3 and clathrin exchange very rapidly at the spindle (Figure 9). In light of this fast exchange, it is perhaps not too surprising that these mesh components do not hang around following MT depolymerization. Finally, related to point 2 above, we are not able to detect mesh in tomograms where there are no MTs, e.g. after depolymerization. This is because our segmentation method works by identifying density that is in contact with MTs. So, while we cannot definitively rule out that the mesh remains after MT depolymerization, we can say that at least this molecular component of the mesh is not resistant to MT depolymerization.

Author response image 2.Fluorescence recovery after photobleaching (FRAP) analysis of TACC3 dynamics in mitotic cells.(A) Average FRAP traces for metaphase cells expressing GFP-tubulin and either mCherry-clathin light chain a or mCherry-TACC3. Plots show the mean ± s.e.m. fluorescence that is scaled to examine the recovery following photobleaching. N = 10 cells. Recovery was best fit by a double exponential function (thick line). (B) Stacked bar chart to show the proportion of recovery by fast (dark) and slow (light) processes. The bar indicates the scaled co-efficient, error bars are ± S.D. of the fitting procedure. Kinetic parameters for the fits are shown below.**DOI:**
http://dx.doi.org/10.7554/eLife.07635.024

We have also added more references to our discussion of the spindle matrix as we had not given due credit to several important papers in the field.

*5) The argument against the preferential binding to the microtubule seam is totally dispensable, since one microtubule is able to bind more than 2 connectors*.

This is true. However, in discussion with colleagues, we are repeatedly asked whether the mesh “footprint” on MTs could be targeting the seam and we think it is important to state our position on this.

*6) How much is TACC3 being overexpressed and what is the effect over the other proteins of the complex? This important control is missing*.

This is now shown in Figure 3—figure supplement 1. Induction of GFP-TACC3 expression more than doubles the total amount of TACC3 in the cells. We have quantified the effect of GFP-TACC3 over-expression previously (Figure 3 in [4]). Overexpression of ch-TOG or of clathrin has no effect on the levels of the other complex members at the spindle. In short, increasing TACC3 expression dials up the amount of the complex on the spindle.